# Effects of the Microbubble Generation Mode on Hydrodynamic Parameters in Gas–Liquid Bubble Columns

**Shanglei Ning, Haibo Jin * , Guangxiang He , Lei Ma, Xiaoyan Guo and Rongyue Zhang**

Beijing Key Laboratory of Fuels Cleaning and Advanced Catalytic Emission Reduction Technology, School of Chemical Engineering, Beijing Institute of Petrochemical Technology, Beijing 102617, China; 2017520021@bipt.edu.cn (S.N.); hgx@bipt.edu.cn (G.H.); malei@bipt.edu.cn (L.M.); guoxiaoyan@bipt.edu.cn (X.G.); zhangrongyue@bipt.edu.cn (R.Z.)

**\*** Correspondence: jinhaibo@bipt.edu.cn; Tel.: +86-010-81292208

**Abstract:** The hydrodynamics parameters of microbubbles in a bubble column were studied in an air–water system with a range of superficial gas velocity from 0.013 to 0.100 m/s using a differential pressure transmitter, double probe optical fiber probe, and electrical resistance tomography (ERT) technique. Two kinds of microbubble generators (foam gun, sintered plate) were used to generate microbubbles in the bubble column with a diameter of 90 mm, and to compare the effects of different foaming methods on the hydrodynamics parameters in the bubble column. The hydrodynamic behavior of the homogeneous regime and the transition regime was also studied. The results show that, by changing the microbubble-generating device, the hydrodynamic parameters in the column are changed, and both microbubble-generating devices can obtain a higher gas holdup and a narrower chord length distribution. When the foam gun is used as the gas distributor, a higher gas holdup and a narrower average bubble chord length can be obtained than when the sintered plate is used as the gas distributor. In addition, under different operating conditions, the relative frequency distribution of the chord length at different radial positions is mainly concentrated in the interval of 0–5 mm, and it is the highest in the center of the column.

**Keywords:** microbubble; foam gun; sintered plate; ERT; optical fiber probe

## 1. Introduction

The bubble column reactor is widely used in gas–liquid reaction systems such as chemical, petrochemical, sewage treatment, mineral processing, and biochemical industries [1–5], due to its simple internal structure, easy operation, and maintenance. For gas–liquid reactions, a large and effective phase contact area is often required, which is more conducive to an increase in the mass and heat transfer rate and thus accelerates the reaction rate. The correct design and operation of these devices depend on the convection type and understanding of global and local properties (gas holdup, bubble chord length distribution). The fluid regime parameters in the bubble column are generally measured by a combination of intrusive and non-intrusive methods, such as differential pressure transmitters [6–8], conductivity probes [9], fiberoptic probes [10], PIV [11], electrical resistance tomography (ERT) [12–14] and photography.

The flow regime in the bubble column can be roughly divided into three stages: homogeneous regime; transition regime and heterogeneous regime [2]. The flow regime is closely related to the superficial gas velocity, bubble column diameter, and initial liquid level height, etc. In addition, the bubble rising velocity and the choice of gas distributor also have a great influence on the flow regime. At a low superficial gas velocity, coalescence and breakup of bubbles rarely occurs; the design of the gas

distributor has a greater influence on the size of the bubbles, which further affects the position of the flow regimen transition point. It is generally believed that, when the superficial gas velocity is less than 0.05 m s$^{-1}$, the size of bubbles generated is 2–10 mm [15]. With the increase in the superficial gas velocity, the flow regime will change. When the superficial gas velocity exceeds the flow regime transition point, the turbulent fluctuations in the column become severe, and coalescence and breakup of bubbles occur, accompanied by changes in the size of the bubbles. Most researchers have studied the effect of the bubble size, superficial gas velocity, and gas–liquid physical characteristics on the hydrodynamic characteristics in the bubble column.

Generally, microbubbles refer to bubbles with a size in the range of 1~1000 μm [16], there are four main types of microbubble generation mechanisms: microfluidics, ultrasound wave, pressure dissolution, and bubble burst due to shear flow/pressure waves [16]. McClure [17] et al. believed that by changing the way of the gas inlet the flow regime can be changed, thus changing gas–liquid mixing in the bubble column. A new type of gas–liquid distributor was used to generate multiple microbubbles by Yang [18] et al. It was found that multiple microbubbles play an important role in improving the mass transfer coefficient, surface concentration and reaction rate constant. It further shows the important role of microbubbles in increasing the reaction rate. Bae [19] et al. studied the generation of microbubbles under high pressure and explored the hydrodynamic characteristics of bubble columns related to microbubble dispersion. Besagni [20] et al. used the needle sparger in the water–monoethylene glycol binary liquid phase. It was observed that with the change in the superficial gas velocity, the curve of gas holdup is "concave", which is different from the "S-shape" produced by the coarse sparger. Hernadez-Alvarado [16] et al. studied the gas holdup, bubble size, and interfacial area in a co-current downflow bubble column with microbubble dispersion by a novel mechanism and found that the new bubble generation technique can get a more high interfacial area compared with conventional bubble columns. The purpose of all research is to understand the flow characteristics of microbubbles in gas–liquid two-phase flow, and then apply it in more extensive fields.

In this study, a microporous sintered plate and a foam gun that can quickly cut the inlet gas into tiny bubbles are used as gas distributors to generate microbubbles in a bubble column with a diameter of 90 mm. The operating superficial gas velocity is 0.013~0.100 ms$^{-1}$. The local average gas holdup, radial gas holdup, and bubble chord length distribution in the column were measured, and the effect of the two foaming methods on the flow regime transition point was further explored by the drift flux method. Therefore, the main goal of this study is to explore the influence of gas distributor design on the hydrodynamic parameters (gas holdup and chord length distribution) and flow regime transition in the column. In turn, this can further expand the application of microbubbles in wastewater treatment and mineral flotation.

## 2. Experimental

The experimental setup is shown in Figure 1. The experiment facility is a Plexiglas® (Germany, Darmstadt) cylindrical bubble column of polymethyl methacrylate with a diameter D = 90 mm, a column height H = 2300 mm, and an initial static liquid level $H_0$ = 1300 mm. The effects of two microbubble generation modes (foam gun and sintered plate) on the hydrodynamic properties of bubble columns were discussed. The sintered plate and the foam gun were used as gas distributors, respectively. The pores of the micropores of the sintered plate were 20 μm, which were fixedly connected to the bubble column by flanges. The air produced by the blower is sent to the bottom air chamber through the flow meter and finally enters the column through the sintered plate. The foam gun is 600 mm long and the diameter of the gas outlet is 1 mm. The foam gun is fixed to the bottom of the column through a polymethyl methacrylate plate. When the air compressor generates 0.35-MPa pressure air, the gas is quickly cut into microbubbles, and it is injected into the column from the outlet of the foam gun. The physical properties of gas and liquid is listed in Table 1.

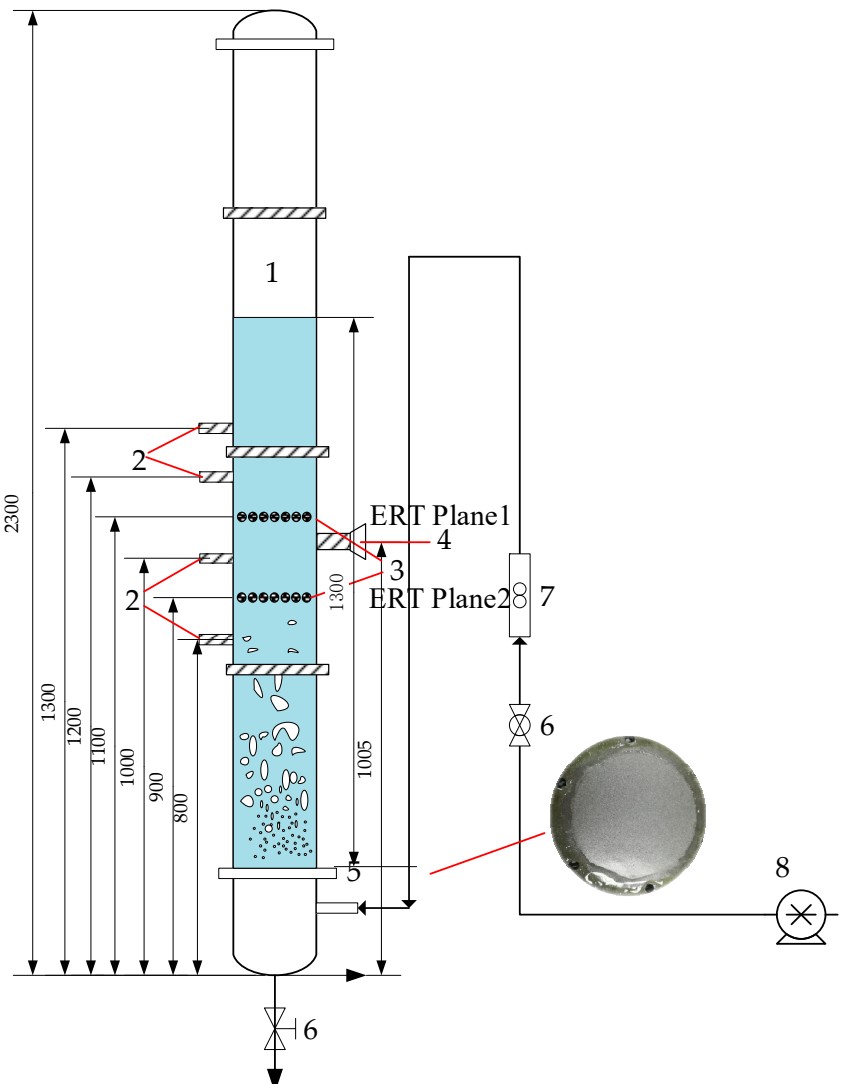

1 Bubble column; 2 Differential pressure transducer; 3 ERT cross section; 4 Fiber optic probe; 5 Gas distributor; 6 Value; 7 Flow meter; 8 Fan; 9 Sintered plate

**Figure 1.** Experimental device diagram of 90-mm bubble column.

**Table 1.** Physical properties of gas–liquid phase.

| Phase | Density/$\rho$ | Viscosity/$\mu$ | Surface Tension/$\sigma$ |
|---|---|---|---|
| Tap Water | 998 kg·m$^{-3}$ | $1.01 \times 10^{-3}$ Pa·s | $7.2 \times 10^{-2}$ N·m$^{-1}$ |
| Air | 1.293 kg·m$^{-3}$ | $17.9 \times 10^{-6}$ Pa·s | |

In this experiment, three measurement methods were used to obtain the hydrodynamic characteristics in the column. The optical fiber probe, the differential pressure method, and the ERT (electrical resistance tomography) were installed in the height range of 800–1300 mm on both sides of the bubble column to get the gas holdup and the chord length.

## 2.1. Differential Pressure Transmitter

The average gas holdup between the two sections is obtained through a differential pressure transmitter. The measurement principle is listed as follows

$$\varepsilon_G = \frac{\Delta P}{(\rho_L - \rho_G)g\Delta H} \approx \frac{\Delta P}{\rho_L g\Delta H} \tag{1}$$

where $\Delta P$ is the pressure difference measured between the two sections, $\rho_L$ is the liquid density, $\rho_G$ is the gas density, $g$ is the acceleration of gravity, and $\Delta H$ is the vertical distance between the two measurement sections.

### 2.2. Fiber Optical Probe

The chord length of bubbles and the local gas holdup at different radial positions are measured by the fiber optical probe.

The fiber optical probe is an intrusive measurement. The principle of optical fiber is that one of the optical fibers serves as the transmitting end of the laser and the other serves as the receiving end of the laser. In the measurement process, when the fiber tip is in the gas phase, the light is completely reflected, with the voltage signal at a high level. When it is in the liquid phase, because the laser is absorbed by the liquid phase, the voltage signal obtained is a low voltage. A series of photoelectric conversions are performed by the acquisition device, and the measured signals are amplified; after the analog-to-digital conversion, the data collection is completed, and the time series of fiber signal changes is obtained. It is precisely because of this difference in refractive index that the optical fiber probe can accurately and in real time measure the hydrodynamic parameters of bubbles in the gas–liquid bubble column. The probe cannot detect the smaller bubbles, so the measured value is lower than the actual experimental value. The differential pressure transmitter is a non-intrusive measurement method, which cannot obtain the local gas holdup, but it is the most accurate and direct method to calculate the average gas holdup between two sections. Therefore, it is necessary to use the measured pressure difference to correct the gas holdup measured by the probe.

The gas holdup is averaged from the data using the fiber optical probe, and then the ratio to the value measured by the differential pressure method is used as the correction coefficient.

As an example, the schematic diagram of bubble parameter measurement is shown in Figure 2. It is measured at five different positions in the radial direction, where $r_1$ is the center of the circle and $r_6$ is the sidewall of the bed, that is, the radius of the bed. The gas holdup and velocity at the sidewall are 0. Six radial positions correspond to gas holdup and bubble velocity.

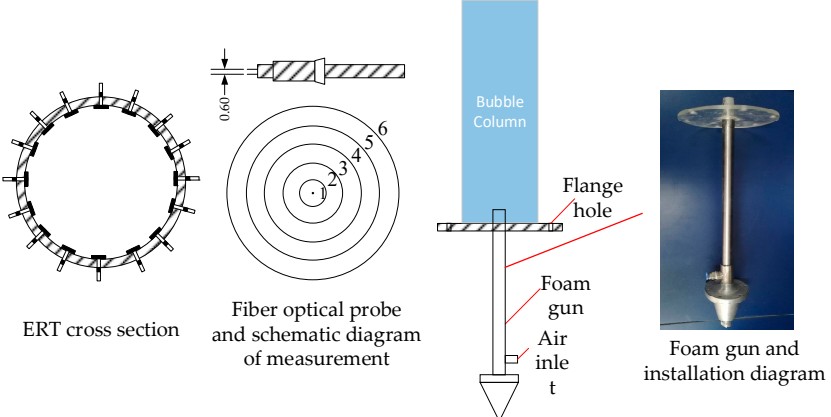

**Figure 2.** Measuring principle and installation method.

The average gas holdup of different rings is added up and divided by the cross-sectional area of the bed to get the average gas holdup of the bed section. The integral expression of the average gas holdup is

$$\overline{\varepsilon} = \frac{2\pi \int_0^R \varepsilon r dr}{\pi R^2} \tag{2}$$

$$\overline{\varepsilon}_i = \frac{(\varepsilon_{i+1} + \varepsilon_i)}{2} \tag{3}$$

$$S_i = \pi\left(r_{i+1}^2 - r_i^2\right) \tag{4}$$

$$\overline{\varepsilon} = \frac{\sum_{i=1}^{5}(\varepsilon_{i+1} + \varepsilon_i)\left(r_{i+1}^2 - r_i^2\right)}{R^2} \tag{5}$$

The velocity of the bubble is based on the time lag between the two sensor output signals $t_0$.

$$U_g = \frac{L}{t_0} \tag{6}$$

where $L$ is the distance between the two probes and $t_0$ is the time for the bubble to pass through the probe.

### 2.3. Electrical Resistance Tomography

Based on the difference in conductivity between the gas and liquid phases, the change of ERT conductivity is transformed into the gas holdup by the Maxwell equation. Finally, a visualization image is used to further represent the changes in the hydrodynamic characteristics of the column. The relationship of Maxwell's transformation is as follows:

$$\varepsilon = \frac{2\lambda_1 + \lambda_2 - 2\lambda_{mc} - \frac{\lambda_{mc}\lambda_2}{\lambda_1}}{\lambda_{mc} - \frac{\lambda_2}{\lambda_1}\lambda_{mc} + 2(\lambda_1 - \lambda_2)} \tag{7}$$

where $\lambda_1$ is the conductivity of the continuous phase (water), $\lambda_2$ is the conductivity of the dispersed phase (air), and $\lambda_{mc}$ is the conductivity value measured during the experiment.

In the gas–liquid phase, the dispersed phase is air, namely, the $\lambda_2$ is 0. The relationship is converted into the following Equation (8).

$$\varepsilon = \frac{2\lambda_1 - 2\lambda_{mc}}{\lambda_{mc} + 2\lambda_1} \tag{8}$$

To verify the consistency of the three measurement results, three measurement methods are used to simultaneously measure the gas holdup under the same operating conditions at the ERT Plane 2, the fiberoptic probe measuring plane, and the differential pressure transmitter under plane 1. The gas holdup measured by the differential pressure transmitter and the ERT is basically consistent. It should be noted that, although the distance between the two probes is already small enough, it may still be unable to detect small bubbles or edge bubbles passing through the probe, so the measured value of the gas holdup is lower than the actual experimental value. The verification results are shown in Figure 3.

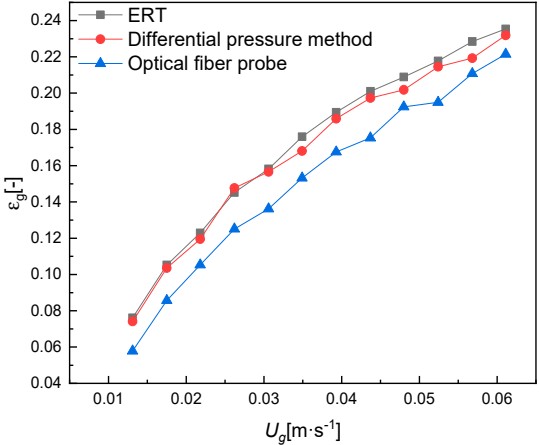

**Figure 3.** The gas holdup $\varepsilon_g$ measured by three methods under different superficial gas velocities.

## 3. Results and Discussion

### 3.1. The Gas Holdup

Microbubbles have a higher specific surface area and a slower movement velocity, which is more advantageous for gas–liquid mass transfer in the bubble column. In the experiment, the superficial gas velocity ranged from 0.013 to 0.100 m/s, and when using a foam gun and a sintered plate as the gas distributors, microbubbles will be generated in the bubble column, and the change trend of gas holdup is similar to the literature [21,22]. As shown in Figure 4, it can be found that the gas holdup increases with the increase in the superficial gas velocity. By comparing the two gas distributors, it can be found that when the sintered plate is used as the gas distributor, the gas holdup is higher before 0.04 m/s and the gas holdup of plane 2 is higher than plane 1, mainly due to plane 2 being closer to the distributor. It can be further concluded that, when a sintered plate is used as the distributor, the axial gas holdup decreases with the increase in H/D, and, at a low superficial gas velocity, the bubbles are small and dense, so the gas holdup is higher. However, as the superficial gas velocity increases, the size of microbubbles gradually increases in the axial position, and the rising velocity of the bubbles increases, so the gas holdup in the measurement plane decreases accordingly. When a foam gun is used as a gas distributor, the gas holdup increases with the increase in the superficial gas velocity, and there is no difference between the gas holdup of plane 1 and plane 2. When the superficial gas velocity increases, the curve of the gas holdup is relatively flat, and there is not much difference between the bubble size and rising velocity.

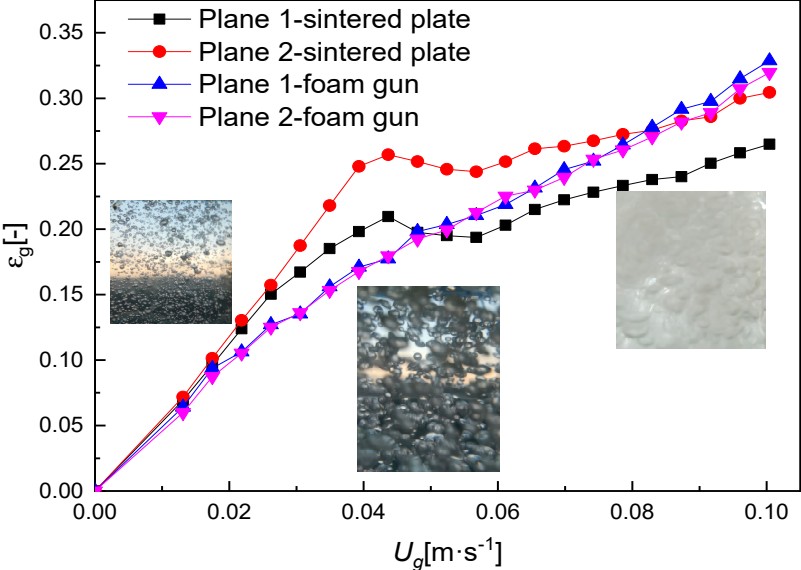

**Figure 4.** Effect of the superficial gas velocity on average gas holdup at different planes.

Figure 5 selects the cross-section gas holdup measured by ERT using two different gas distributors (sintered plate and foam gun) under the superficial gas velocities 0.026 m/s, 0.052 m/s, 0.079 m/s and 0.100 m/s. It can be seen that the foam gun as a gas distributor has a significantly higher gas holdup than the sintered plate, and the cross-section gas holdup in the column is evenly distributed at a low superficial gas velocity. In the same cross-section, the gas holdup gradually decreases from the center along with the radial position and is symmetrically distributed. Compared with traditional measuring methods, ERT expresses the change in the gas holdup in the column intuitively and visually, which is more conducive to understanding the flow in the bubble column.

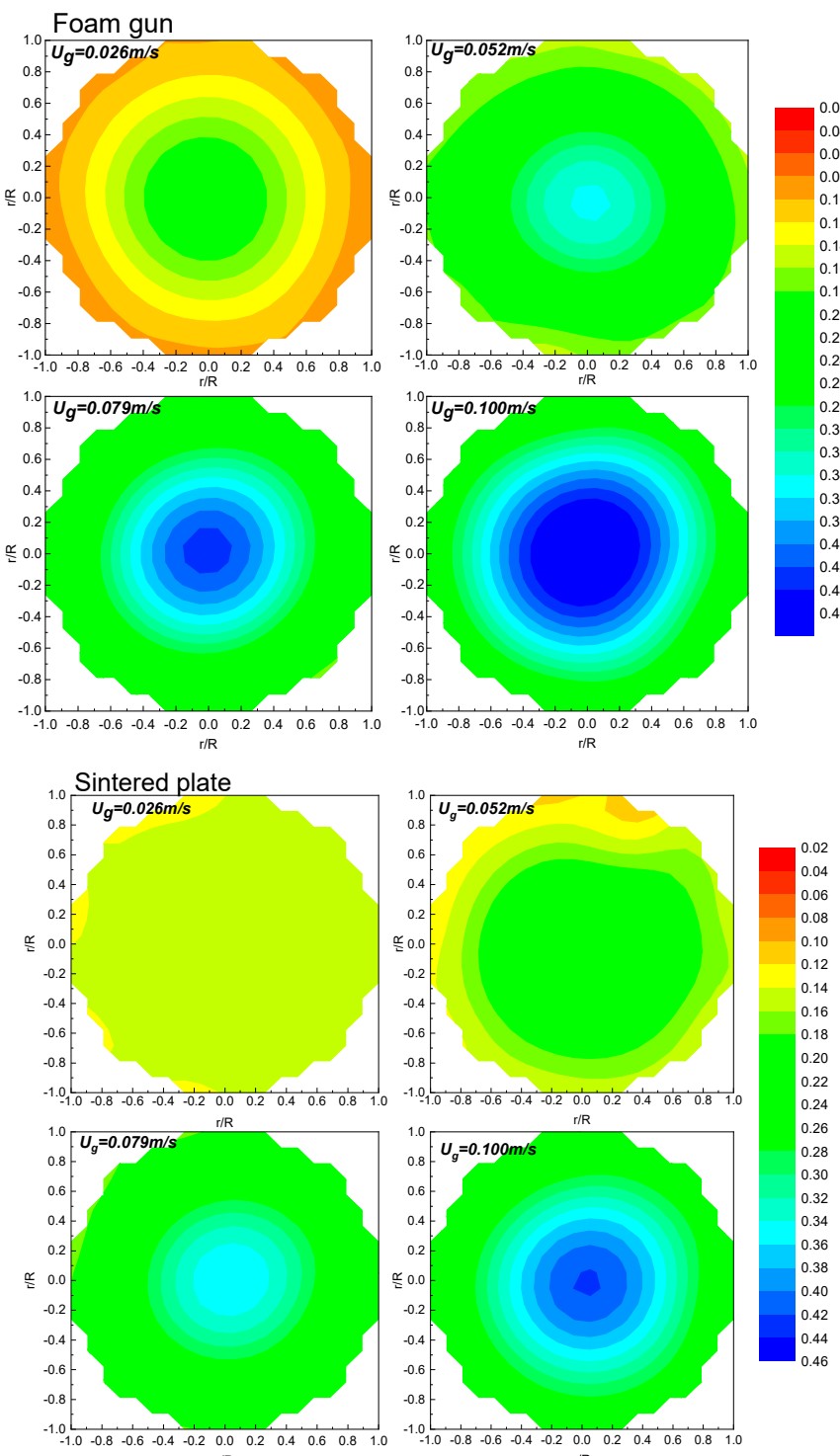

**Figure 5.** The cross-section distribution of gas holdup measured by ERT at different superficial gas velocities.

## 3.2. Drift Flux Model

The flow regime in the bubble column is generally divided into the homogeneous regime, transition regime, and heterogeneous regime. For the microbubble generator, when the superficial gas velocity increases to a certain value, the interaction between the microbubbles is enhanced, and microbubbles will coalescence into small bubbles. In addition, the rising velocity of bubbles becomes faster. As the superficial gas velocity and gas holdup increase, the flow regime begins to

change from the homogeneous regime to the transition regime. It is of great significance for the design and application of the bubble column to accurately identify the flow pattern in the bubble column. The flow regime can be roughly determined by $\varepsilon_g - U_g$ mapping, and the result is shown in Figure 4. Most researchers usually use the drift flux model proposed by Zuber and Findlay [23], as shown in Equation (9):

$$\frac{U_g}{\varepsilon_g} = C_0 (U_g + U_l) + C_1 \tag{9}$$

Since the bubble column is operated in batch mode, the superficial liquid velocity is 0. So Equation (9) is converted to Equation (10).

$$\frac{U_g}{\varepsilon_g} = C_0 \cdot U_g + C_1 \tag{10}$$

In Equation (10), $C_0$ is a parameter representing the radial uniform distribution of gas holdup and velocity; when $C_0$ is close to 1, the radial distribution of gas holdup is the most uniform. Therefore, the flow pattern can be determined by the change in $C_0$. Figure 6 shows the relationship between the drift flux given by Equation (10) and the superficial gas velocity for the sintered plate system and foam gun system. When the sintered plate was used as the gas distributor, the change of the gas–liquid flow with the operating superficial gas velocity was greater, and the changes on the two planes were inconsistent, which is consistent with the conclusion in Figure 4. For the foam gun, the flow pattern in the column can also be obtained by the drift flux method. The flow pattern changes at a higher superficial gas velocity, and the state in the uniform bubble regime is more stable, which is conducive to the relevant gas–liquid reactions in the microbubble state.

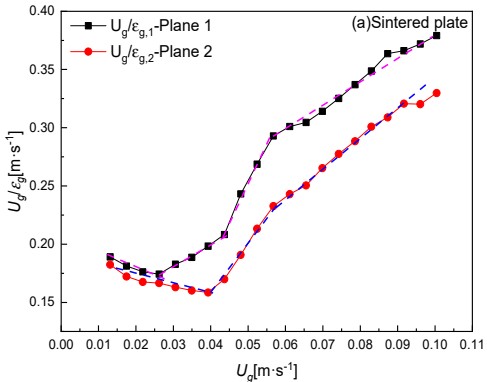 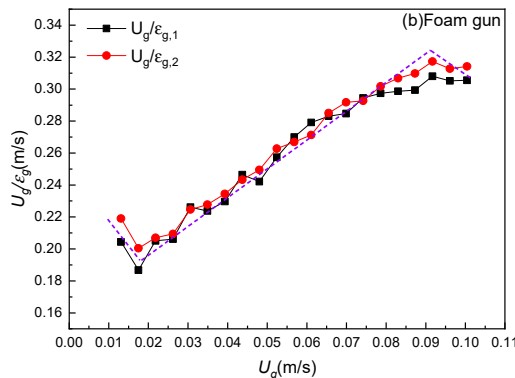

**Figure 6.** Flow pattern identification with the drift flux method: (**a**) sintered plate and (**b**) foam gun.

In most cases, the parameters of $C_0$ and $C_1$ are not directly reported in the literature, because it is difficult to measure the velocity of the liquid phase in two-phase flow. Therefore, Wallis [24] proposed another drift flux theory to judge the flow pattern. Wallis [24] modified the drift flux model to Equation (11).

$$U_w = U_g (1 - \varepsilon_g)^{n-1} \pm U_l \varepsilon_g \tag{11}$$

where $U_w$ is defined as the drift flux velocity, and $n$ is the Richardson–Zaki index, with a value of 2.39 for low viscosity (high Reynolds number) systems [25].

$U_l$ is 0 in the batch mode, and the value of $(n-1)$ is close to 1 in a low viscosity fluid, so Equation (11) becomes Equation (12):

$$U_w = U_g (1 - \varepsilon_g) \tag{12}$$

Figure 7 shows the relationship between the drift flux given by Equation (12) and the gas holdup in the sintered plate system. Because the drift flux is only stable in the uniform regime, the critical point of the slope change is the transition point of the flow regime. Therefore, the drift flux theory

can determine the transition point, and can also obtain the corresponding gas holdup when the flow pattern changes, which is also important for the delicate gas–liquid reaction.

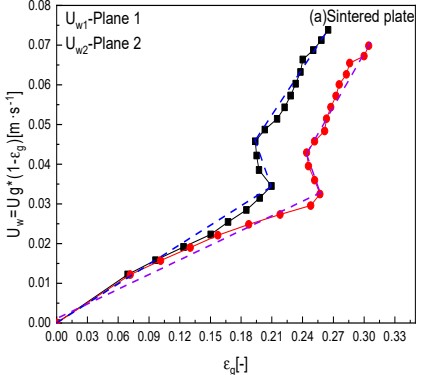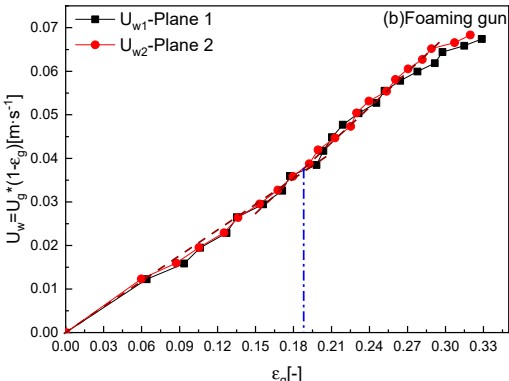

**Figure 7.** Effect of different gas distributors on flow regime transition with an air–water system: (**a**) sintered plate and (**b**) foam gun.

Compared with Figure 4, the curve of the drift flux velocity with the gas holdup is basically the same as the curve of the gas holdup with the superficial gas velocity. Furthermore, the mapping drift flux velocity $U_w$ vs. $\varepsilon_g$ is more accurate than $U_g/\varepsilon_g$ vs. $U_g$, which is of great significance to understand the gas–liquid flow in the column.

### 3.3. Bubble Chord Distribution

It can be seen from the above experimental results that when the foam gun is used as a microbubble generation, the gas holdup in the bubble column is higher and the foaming effect is better. Therefore, it is of great significance to study the effect of the foam gun used as a microbubble generation mode on the bubble chord length in the column. With a sintered plate and a foam gun as the gas distributor, the mean bubble chord length with the superficial gas velocity at different radial positions is shown using an optical fiber probe in Figure 8. Obviously, under the same superficial gas velocity, the mean bubble chord length at different radial positions is inconsistent, but within the range of operating gas velocity, the distribution of mean chord length is basically similar, between 0.006 m–0.012 m, and the bubble size distribution is narrow. By comparing the two kinds of microbubble generator, when the foam gun is used as a gas generation device, the minimum mean chord length is about 0.004 m, which is smaller than that produced by the sintered plate.

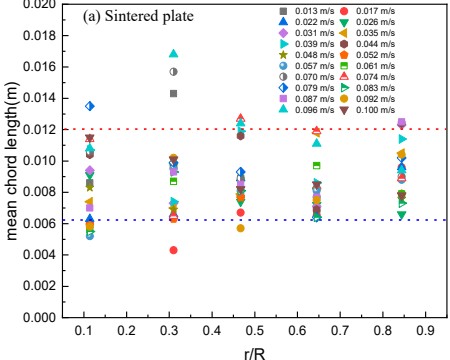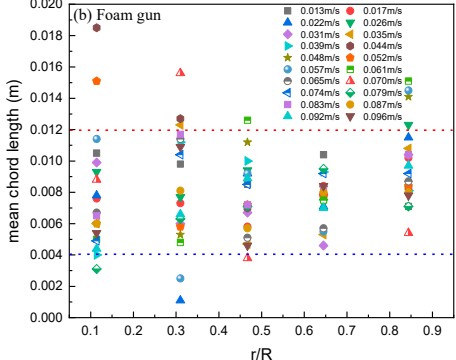

**Figure 8.** Mean chord length measured at different radial positions when using different generation devices: (**a**) sintered plate and (**b**) foam gun.

Figure 9 shows the relative frequency distribution of the chord length using a foam gun as the microbubble generation device, and the chord length relative frequency distribution at different radial positions varies with the superficial gas velocity. It can be seen from Figure 9 that, under three different

superficial gas velocities and different radial positions, the relative frequency distribution value is maximum when the chord length is 2 mm; the relative frequency distribution of the chord length of 0–5 mm accounts for the majority of this, and it also indicates that the bubble chord length is narrow even when the fluid in the column has a different flow regime. Guan [7] used a perforated plate with 73 holes of 1.5 mm diameter as gas distributors. The maximum gas holdup measured under the same superficial gas velocity was 0.15, and most of the bubble chord lengths are distributed in the interval of 3–7 mm. When the probe is located at different radial positions, the bubble chord length peak of the relative frequency distribution does not move significantly, which shows that the bubble size distribution of the entire system is uniform. For the location r/R = 0.11 and r/R = 0.31, the peak of the relative frequency distribution is higher, indicating that the bubbles are smaller and closer to the center of the column.

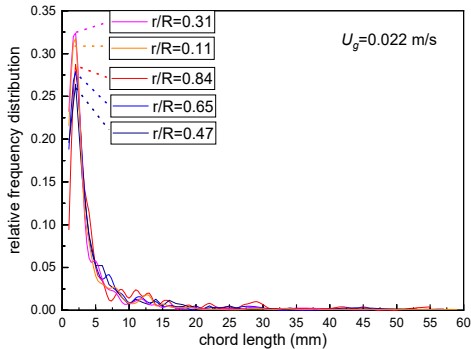 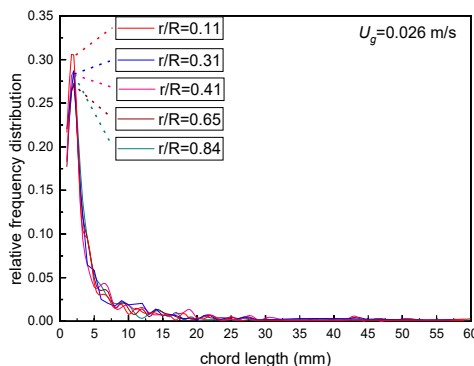

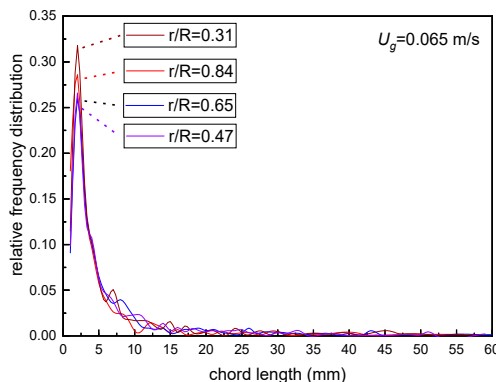

**Figure 9.** The relative frequency distribution of chord length as a function of radial position at different superficial gas velocities.

### 3.4. Radial Gas Holdup

Fiberoptic probes are used to measure the radial gas holdup distribution of different microbubble generators in the operating condition; the result is shown in Figure 10. Figure 10a is the measurement result when the sintered plate is used as the gas distributor and Figure 10b is the measurement result when the foam gun is used as the distributor. With the increase in the superficial gas velocity, the radial gas holdup increases, and the gas holdup gradually decreases from the center of the bubble column to the wall, which is consistent with most experimental results [20,26]. At the same time, under the same superficial gas velocity, the radial gas holdup of the foam gun distributor is higher than that of the sintered plate.

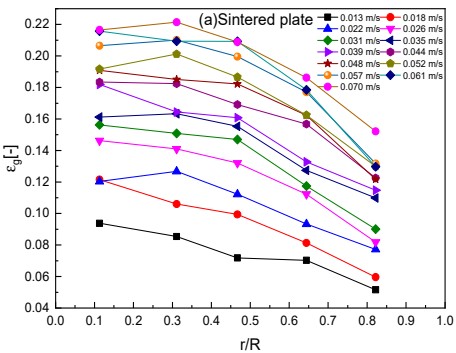 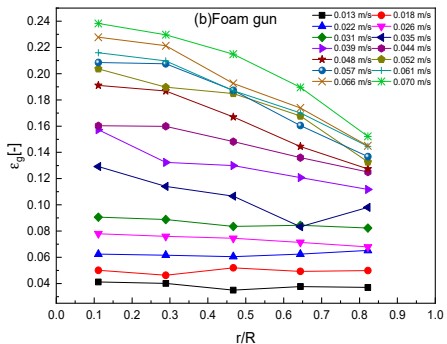

**Figure 10.** Radial gas holdup measured by the fiberoptic probe. (**a**) Sintered plate. (**b**) Foam gun.

## 4. Conclusions

In this paper, sintered plates and foam guns were used as gas distributors to measure the hydrodynamic characteristics in the bubble column. It is very important for the reaction of microbubbles. The experimental results are compared with relevant literature, the following conclusions are obtained:

(1) Sintered plate and foam gun are good microbubble generating devices. At the same operating gas velocity, the gas holdup of the two distributors is significantly higher than conventional distributors. For gas–liquid reactions such as sewage treatment, the reaction rate can be increased.

(2) Compared with the sintered plate, the gas holdup the foam gun produces is higher at the operating gas velocity, and it is in the uniform bubbling regime for a long time. According to the image measured by ERT, the gas holdup in the center of the column is the highest, the gas holdup at the wall is 0, and the distribution is almost uniform.

(3) In generation devices, the bubble gun is used as the gas distributor to generate a narrow chord length and relative frequency distribution. At different radial positions, the peak position of the chord length of the bubble does not change. When the probe position is close to the center of the bubble column, it can get a higher peak value.

(4) Based on the measurement data of the two types of distributors, the drift flux model was studied, and the flow pattern transition was determined, which provides a more favorable basis for the accurate understanding of the fluid flow in the column.

**Author Contributions:** Investigation, S.N., G.H., L.M.; resources, S.N., X.G., R.Z.; data curation, S.N., G.H.; writing—original draft preparation, S.N., G.H., L.M., X.G., R.Z., H.J.; writing—review and editing, S.N., H.J.; supervision, H.J.; project administration, H.J. All authors have read and agreed to the published version of the manuscript.

**Funding:** This research was funded by the National Natural Science Foundation of China, grant number 91634101, and The Project of Construction of Innovative Teams and Teacher Career Development for Universities and Colleges under Beijing Municipality, grant number IDHT20180508.

**Conflicts of Interest:** The authors declare no conflict of interest.

## Nomenclature

| | | |
|---|---|---|
| $\Delta H$ | [m] | Vertical distance between the two measurement sections. |
| $\Delta P$ | [Pa] | Pressure difference measured between the two sections |
| $L$ | [m] | Distance between the two probes, m |
| $t_0$ | [s] | Time for the bubble to pass through the probe, s |
| $\lambda_1$ | [mS cm$^{-1}$] | Conductivity of the continuous phase (water), mS/cm |
| $\lambda_2$ | [mS cm$^{-1}$] | Conductivity of the dispersed phase (air), mS/cm |
| $\lambda_{mc}$ | [mS cm$^{-1}$] | Conductivity value measured during the experiment, mS/cm |
| $U_i$ | [m s$^{-1}$] | Velocity, m·s$^{-1}$, g = gas phase, l = liquid phase |
| $U_w$ | [m s$^{-1}$] | Drift flux velocity, m·s$^{-1}$ |
| $H/D$ | [–] | Height to column diameter ratio |
| $C_0$ | [–] | drift-flux coefficient |

| | | |
|---|---|---|
| $C_1$ | [m s$^{-1}$] | drift-flux coefficient, m·s$^{-1}$ |
| $n$ | [–] | Richardson-Zaki index |
| $R$ | [m] | Diameter of the measuring plane |
| $r/R$ | [–] | Radial location |
| $\varepsilon_g$ | [–] | Gas holdup |
| $\varepsilon$ | [–] | Average gas holdup |
| $g$ | [m s$^{-2}$] | Gravitational acceleration, m·s$^{-2}$ |
| $\rho$ | [kg m$^{-3}$] | Density, kg·m$^{-3}$ |
| $\mu$ | [Pa s] | Viscosity, Pa·s |
| $\sigma$ | [N m$^{-1}$] | Surface tension, N·m$^{-1}$ |

## Lower Subscript

| | |
|---|---|
| $G$ | gas phase |
| L | liquid phase |
| $i$ | referring to the gas phase or the liquid phase |

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
