# Peer review of "Effects of the Microbubble Generation Mode on Hydrodynamic Parameters in Gas–Liquid Bubble Columns"

_processes, doi:10.3390/pr8060663_

Round 1

Reviewer 1 Report

This is an interesting paper, demonstrating control of the size of bubbles and gas hold-up in columns and dependence of the parameters on the bubble formation mechanism. The experiments and the way they have been analysed are clearly described, and the results are clearly presented.

One or two minor points should be corrected:
Line 59 refers to Hernadez-Alvarado, rather than Hernandez-Alvarado.
The notation used in lines 102-103 (R1 etc) differs from that in the equations (ri).
In equation 5, the sum over i should run from 1 to 5, not 1 to 6.

In the conclusions, points 2 to 4 are fully supported by the results in the paper. Point 1, however, states “At the same operating gas velocity, the gas holdup of the two distributors is significantly higher than conventional distributors.” It would be helpful for the reader if the introduction to the paper included some typical velocity and hold-up values from the literature referred, to provide justification for this claim.

This paper reports interesting work, and it deserves publication, but the authors should provide more justification for the first of their conclusions.

Reviewer 2 Report

There is a lack of details of the bubble generator, a lack of description of the measuring and measuring equipment, and in particular of a description of matters relating to the accuracy of the measuring equipment.

I thought these were very important for the  applicability of the formulas used in the comparison.

The most ambiguous point is whether or not micro-order bubbles are present.

Of course, it can be understood by estimating from the bubble velocity, 

but it is difficult to understand without a photograph.

(1) If possible,  photograpgh of microbubbles should be shown.

(2) Please add in text the dimensions of foam gun and sintered palte.

Also,  if possible, please illustrate how to set up foam guns in bubble column .

(3)As shown in the experimental device (fig.1), I did not understand why  authors measured  at the position where coalescence occurs. Why did Authors chose the measurering position?

(4) It is necessary to reconfirm the numbers in Figure6 (textp6, line175).

(5) Small, unclear characters in Fig.4 need to be rewritten.

Please reconfirm  significant figures in colar bar.

(6)How much differences between the average gas holdup (page3, eq .(1) and average values of ERT (page4, eq. (7)) or fiber probe (page 3, line 91) ?

Reviewer 3 Report

The authors present an experimental study measuring gas hold-up, with different techniques, in bubble columns for two micro-bubble generating devices: sintered plates and foaming guns. They show that foaming guns produce larger gas holdup than sintered plates in the same operating conditions, with a more homogenous bubble size distribution.

I cannot recommend publication in the manuscript in its current form for three main reasons. First, I fail to see what was the question addressed by the current study by reading the introduction. A few papers are cited, but with no relation to the present study, and only a rough description of the study is given, but the purpose remains unclear in the introduction.

Second, the description of the setup and measuring techniques is a bit quick, without any concrete details. This is especially the case for the fiber optics probe, where the range of measured diameter is just discarded to the sentence “The probe cannot detect the smaller bubbles, so the measured value is lower than the actual experimental value.”, with no reference or explanation.

Third, I fail to see the contribution made by the paper (corelated to concern 1). If a model from 1965 captures the data almost perfectly, what is the point of the study (see fig 5)? The only clear comparison to the current data with the literature I notice was “the trend of the gas holdup is similar to the literature [22,23].” Besides, the effect of superficial gas velocity seems to have a much larger impact than the injection method…

Some other comments:

  • In the introduction, it would be nice to give a order of magnitude of what small and smaller means, for broader audience sake (line 33). Also what is a normal bubble size? (line 38)
  • Line 52 reword the first sentence.
  • Line 65 the author should give some context, because I would deem a 1000µm bubble millimetric.
  • Line 76 cap “the”
  • Line 44 can you be quantitative?
  • Line 47 fig 4 not 3

Round 2

Reviewer 3 Report

My concern with point (1) was rather a major one, and I am afraid the authors eluding answering the question, and only made minor modifications in the introduction. A reader is still not able to understand how this study is situated compared to existing articles, what questions are being addressed in the present study and what is its originality.

The answer to point (4) was elusive too, and lacked quantitative arguments.

The authors answer my other points and made modifications. I think the new curve for Fig. 3 with the optical probe deserves a better justification of why it lies below the two other curves with a constant shift over the range of velocities (only why it lies below is addressed).
